# Thermosensitive PBP2a requires extracellular folding factors PrsA and HtrA1 for *Staphylococcus aureus* MRSA β-lactam resistance

Mélanie Roch[1], Emmanuelle Lelong[2], Olesya O. Panasenko[1,2], Roberto Sierra[1], Adriana Renzoni[2] & William L. Kelley [1]*

*Staphylococcus aureus* is a major human pathogen and represents a clinical challenge because of widespread antibiotic resistance. Methicillin resistant *Staphylococcus aureus* (MRSA) is particularly problematic and originates by the horizontal acquisition of *mecA* encoding PBP2a, an extracellular membrane anchored transpeptidase, which confers resistance to β-lactam antibiotics by allosteric gating of its active site channel. Herein, we show that dual disruption of PrsA, a lipoprotein chaperone displaying anti-aggregation activity, together with HtrA1, a membrane anchored chaperone/serine protease, resulted in severe and synergistic attenuation of PBP2a folding that restores sensitivity to β-lactams such as oxacillin. Purified PBP2a has a pronounced unfolding transition initiating at physiological temperatures that leads to irreversible precipitation and complete loss of activity. The concordance of genetic and biochemical data highlights the necessity for extracellular protein folding factors governing MRSA β-lactam resistance. Targeting the PBP2a folding pathway represents a particularly attractive adjuvant strategy to combat antibiotic resistance.

[1] Department of Microbiology and Molecular Medicine, University Hospital and Medical School of Geneva, 1 rue Michel-Servet, Geneva CH-1211, Switzerland. [2] Service of Infectious Diseases and Department of Medical Specialties, University Hospital and Medical School of Geneva, 4 rue Gabrielle-Perret-Gentil, Geneva CH-1206, Switzerland. *email: William.Kelley@unige.ch

Staphylococcus aureus is a major human pathogen causing mild to life-threatening infections and yet it is a commensal organism transiently colonizing 20–30% of the world's population[1]. MRSA strains originate by horizontal acquisition of an SCCmec cassette encoding a variant DD-transpeptidase, PBP2a, belonging to the high molecular weight Class B1 penicillin binding protein (PBP) family[2,3]. PBP2a renders MRSA strains resistant to virtually all β-lactam antibiotics and for this reason S. aureus is considered an ESKAPE organism highlighted as an urgent research priority[4–6]. Antibiotic multi-resistance, or cross-resistance, renders the clinical management of MRSA infections particularly problematic[7–9].

MRSA β-lactam resistance is explained by the discovery that PBP2a is allosterically regulated[10,11]. The active site serine in the catalytic domain resides within a narrow channel and its opening is gated by binding of peptidoglycan to the N-terminal domain over 60 Å away. The success of the last generation cephalosporins such as ceftaroline occurs because ceftaroline can bind the allosteric site and productively trigger active site gating[10]. Other β-lactams fail to trigger allostery, thus accounting for the broad resistance mechanism[12,13]. Since the last two steps of peptidoglycan biosynthesis, transglycosylation and transpeptidation occur on the outer leaflet of the cell membrane[14], PBP2a must be vectorially translocated by the general secretion apparatus as an unfolded polypeptide, and then somehow acquire its functional membrane-anchored tertiary configuration. A protein transported across the membrane in an unfolded state could conceivably fold spontaneously, or it could interact transiently with protein folding chaperones that prevent unfavorable off-pathway aggregation and favor progression to a proper final folded state. An efficient quality control system should assure that improperly folded proteins are identified and degraded.

The HtrA family of dual protein chaperone/serine proteases plays an essential role in the quality control of secreted proteins in many organisms[15]. In prokaryotes, HtrA proteins are associated with the proper expression of exoproteins and mitigating secretion stress. The prototypical HtrA protein is a homotrimer with each polypeptide displaying a serine protease domain and at least one PDZ domain thought to limit HtrA protease activity to specific client proteins[16].

HtrA family proteins can also cooperate with protein chaperone peptidyl-prolyl isomerases (PPIases) in both Gram-negative and Gram-positive organisms[17–20]. In previous work, we found that the lipoprotein PrsA, a member of the parvulin family of PPIases[21,22] was non-essential in S. aureus, unlike Bacillus subtilis where prsA is essential except in presence of high levels of magnesium[20]. Depletion of PrsA in B. subtilis is associated with secretion stress and is thought to impact directly or indirectly certain PBPs affecting lateral cell wall biosynthesis[20]. Consistent with this observation, our work showed that S. aureus PrsA disruption caused slight changes in β-lactam resistance, as well as reduced PBP2a levels in membrane extracts[21,22].

We hypothesized that PBP2a folding by extracellular chaperones such as PrsA and HtrA assures the quality control necessary for MRSA to display broad β-lactam resistance. Herein, we report that purified PBP2a displays a pronounced propensity for protein misfolding at physiological temperatures and unfolding was dramatically enhanced with minor shift to acidic conditions. Unfolding transitions in PBP2a, resulted in aggregation and activity loss. Genetic experiments showed that a double mutant htrA1/prsA resulted in synergistic re-sensitization to β-lactam antibiotics in the model MRSA strain COL and that this effect was particularly apparent with minor increase in temperature. Evidence presented shows that PrsA provides a protein chaperone function to control aggregation and HtrA1 ensures proteolytic degradation of misfolded PBP2a. Since clearly PBP2a is

thermosensitive and requires a complex folding architecture to accomplish allostery, we conclude that HtrA1 and PrsA are necessary to assure translocational folding quality control. Our discovery of a catastrophic folding trap of PBP2a suggests novel avenues to pursue for development of adjuvant small molecules that target the intrinsic folding process of the enzyme, or compromise the extracellular folding quality control system.

## Results

**Protein unfolding properties of purified PBP2a.** We reasoned that PBP2a has off-pathway folding conformers that are inactive and would compromise function if not removed by a dedicated quality control system involving protein chaperones subsequent to translocation. To provide evidence of misfolding propensity, we first determined whether purified PBP2a showed detectable unfolding transitions affecting its activity.

We purified recombinant PBP2a and showed that it was correctly folded using covalent active site serine derivatization by bocillin FL (a fluorescent penicillin V that can label PBP2a and other PBPs[23]) and subject to allosteric regulation in a competition assay using pre-exposure to an anti-MRSA β-lactam, ceftaroline (which can trigger allostery and acylate the active site), or oxacillin, an ineffective β-lactam (which cannot trigger allostery) (Supplementary Fig. 1; uncut gels are shown in Supplementary Fig. 2). The complexity of the salt bridge network[12] required for the allosteric mechanism serves as an excellent indicator of correct protein folding.

Thermal aggregation assays were performed by incubating PBP2a at defined temperatures and pH followed by separation into soluble and pellet fractions. The results showed a clear midpoint onset of insolubility at 45–46 °C at pH 7.4 (Fig. 1a; uncut gels are shown in Supplementary Fig. 3). The external membrane face of gram-positive organisms lacks a true periplasmic space folding compartment and resembles a strong anion exchange resin[24]. An acidic pH gradient near the outer membrane leaflet would also be expected because of the proton gradient contribution to the membrane potential[25,26]. To test whether minor changes in pH also affected PBP2a solubility, we performed thermal aggregation assays at pH 7.0, 6.6, 6.2, and 5.8 and at temperatures ranging from 37 to 46 °C. Our results clearly demonstrated that purified PBP2a has a dramatic tendency to precipitate over a narrow temperature range that was further exacerbated by mild acidic conditions (Fig. 1b). Since purified PBP2a that had never been frozen did not precipitate at 5 mg mL$^{-1}$ at 4 °C for at least five months in neutral pH buffer nor display lack of its ability in bocillin-FL competition assays over time (Supplementary Fig. 1C), we conclude that elevated temperature and mildly acidic pH shift results in PBP2a precipitation.

In order to determine whether PBP2a in the pellet fraction was functional, we performed bocillin-FL labeling and competition assays. Our results showed that insoluble PBP2a pellets could not be efficiently derivatized by bocillin FL and did not display allostery indicating that they were completely inactive (Fig. 1c).

To refine our analysis of the early onset of unfolding, we performed differential scanning fluorimetry using the protein-specific hydrophobic fluorescent probe Sypro orange[27]. Our results showed a sharp unfolding transition at pH 7.4 with a detectable onset at 36 °C and maximum fluorescence rate of change (melting point) at 45 °C. Above this temperature, fluorescence decreased consistent with aggregation and probe displacement. When the pH was shifted to 5.8, we found an identical curve profile, but the unfolding transition onset shifted 3 °C lower (Fig. 1d). These results indicate that PBP2a exposes hydrophobic regions prior to the onset of irreversible aggregation and precipitation. Importantly, the temperature range of

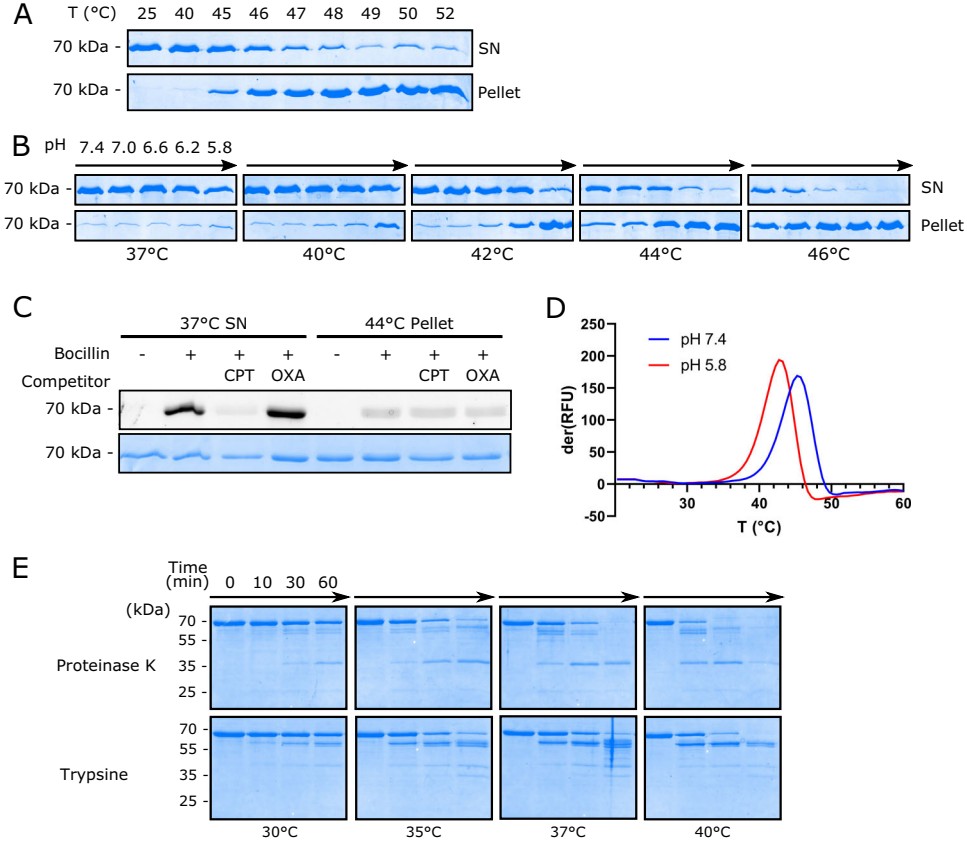

**Fig. 1 a** Thermal aggregation assay of PBP2a performed at pH 7.4 over the temperature range 25–52 °C. (SN: soluble supernatant fraction; pellet: insoluble fraction). Note the strong onset of insoluble PBP2a in the 40–46 °C interval and consistent with the fluorimetric assay in panel **d**. **b** Thermal aggregation assay of PBP2a conducted at various temperatures and in pH ranges from 5.8 to 7.4. Note that PBP2a precipitation increases with both increasing temperature and decreasing pH. **c** PBP2a activity assay using bocillin-FL active site derivatization and allosteric competition with ceftaroline (CPT) or oxacillin (OXA). For competition assays, PBP2a was pre-incubated with excess CPT or OXA prior to the addition of bocillin FL. Bocillin fluorescence was recorded by fluorescence emission of unstained SDS gels then stained. PBP2a in the 37 °C supernatant remains fully active, while PBP2a recovered from the 44 °C pellet has lost activity. **d** Differential scanning fluorimetry of PBP2a with Sypro Orange performed at pH 7.4 (blue), or pH 5.8 (red). The first derivative plot shows relative fluorescence units change (derRFU) with temperature. The onset of hydrophobic residue exposure indicative of the initiation of unfolding begins to be detectable at 36 °C at pH 7.4, and at 33 °C at pH 5.8. Peak fluorescence change occurs at 46 °C at pH 7.4 and 43 °C at pH 5.8 after which protein aggregation and loss of signal progressively occurs. **e** Partial proteolysis of PBP2a using either Proteinase K or trypsin at the indicated time and incubation temperature. Note the appearance of protease-resistant fragments and cleavage sensitivity consistent with altered protein folding. Uncut gels are shown in Supplementary Fig. 3

unfolding occurs in a physiologically relevant window: >36 °C and within a pH range that would be expected on the outer leaflet of the plasma membrane.

We also performed partial proteolysis assays to detect early stage unfolding conformers. We observed considerable changes in sensitivity to mild proteolysis by both trypsin and proteinase K over the temperature range 30–40 °C suggesting the onset of protein unfolding transitions that were susceptible to proteolysis (Fig. 1e). Western blot analysis showed that both proteases rapidly cleaved the N-terminal allosteric domain regions of PBP2a explaining the appearance of prominent smaller bands migrating with apparent molecular weights in the range 7–12 kDa less than native PBP2a (Supplementary Fig. 4). When PBP2a was pre-derivatized with bocillin FL prior to proteolysis, we also discovered that the catalytic domain was particularly resistant to trypsin or proteinase K proteolysis, whereas the N-terminal domain rapidly disappeared (Supplementary Fig. 4).

Collectively, these results show that PBP2a possesses a catastrophic folding transition near physiological conditions that abolishes its activity. PBP2a also possesses protease sensitivity in the allosteric domain that can be further sensitized by mild increased temperature. In light of these findings, we turned to a

deeper genetic study. We reasoned that extracellular protein folding factors such as PrsA and HtrA were important for the post-translocational folding of PBP2a in order to avoid such off-pathway and deleterious misfolding. To this end, we examined the properties of *prsA* and *htrA1* mutants alone, or in combination.

**Disruption of both *prsA*/*htrA1* affects β-lactam resistance.** *S. aureus* encodes two HtrA family members: *htrA1*, and *htrA2* (Fig. 2a; Supplementary Fig. 5A). Both display a tri-domain signature: a transmembrane domain (TM) TLSP (trypsin-like serine protease) and C-terminal PDZ domain. Both HtrA1 and HtrA2 are detected in *S. aureus*, yet only HtrA1 resides on the cell surface[28]. The large N-terminal domain of unknown function in HtrA2 probably alters its cellular localization and topology. Both *prsA* and *htrA1*, but not *htrA2*, are induced by cell wall active antibiotic stress and are controlled by the VraTSR two-component sentinel system[29] (Supplementary Fig. 6 and Supplementary Table 1). The coordinate expression of *prsA* and *htrA1* in response to cell wall stress[30,31] suggests a need for these extracellular protein folding factors in these conditions, and is

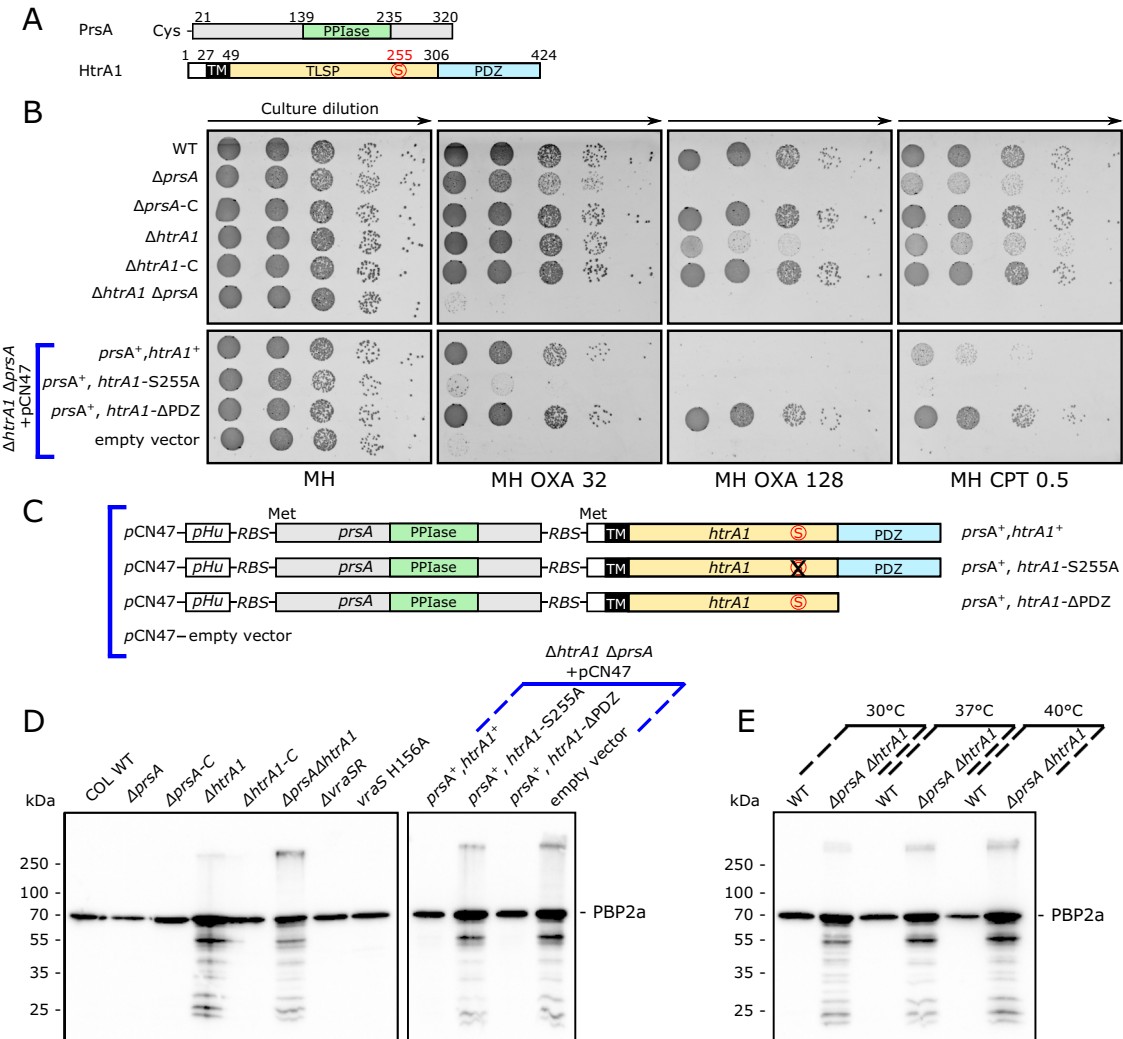

**Fig. 2 a** Schematic diagram of *S. aureus* PrsA, and HtrA1. Numbering refers to amino acid codon. The PrsA parvulin-like PPIase domain is shown and also a lipoprotein signal sequence and lipidation at Cys21. HtrA1 is shown with transmembrane region (TM), trypsin-like serine protease domain (TLSP) with active site serine of the catalytic triad shown in *red*, and the C-terminal PDZ domain. **b** Colony forming unit (cfu) assay on cation-adjusted Mueller-Hinton (MH) agar plates with the indicated antibiotics (OXA:oxacillin; CPT: ceftaroline) and the concentration in µg mL[−1]. Aliquots (10 µL) of serial 10-fold dilutions (arrows) were spotted. *Upper panel* shows COL wild-type (wt) or the indicated mutant derivatives. *Lower panel* depicts the complementation of the *prsA/htrA1* double mutant with pCN47 vector alone or the indicated dual operon. **c** Schematic of the pCN47 plasmids used for complementation of Δ*prsA*/ Δ*htrA1* showing the hybrid operon design strategy. **d** Western blot of membrane extracts from the indicated cells using anti-PBP2a antibody. Degradation products appear only when *htrA1* is disrupted. The *right panel* shows complementation of the *prsA/htrA1* mutant with the indicated pCN47 plasmids. An active HtrA1 protease is necessary to restore quality control since the HtrA1 active site S255A mutant fails to remove PBP2a degradation fragments. **e** Western blot using anti-PBP2 antibody and membrane extracts prepared from cells grown at the indicated temperatures. Note the decrease in full-length PBP2a with increasing temperature in wild-type cells compared with increased degradation fragments and accumulation of full-length PBP2a in *htrA1* mutant cells. Uncut membranes of **d**, **e** are shown in Supplementary Fig. 10

consistent with their cooperation in the secretion process in other organisms[17–20]. Both PrsA and HtrA1 are also strongly induced when signal peptidase is inhibited by arylomycin leading to secretion stress[32].

We made all possible single, pairwise and triple combinations of disruptions in *htrA1*, *htrA2*, and *prsA* in the model MRSA strain COL. The resulting strains were first tested for antibiotic sensitivity by colony forming assay and microdilution MIC (minimal inhibitory concentration) assay. The results are shown in Table 1 and Fig. 2b and Supplementary Fig. 5B).

Disruption of *htrA2* showed no effect alone, nor in combination with any other mutations tested (Supplementary Fig. 5B). In contrast, we observed striking evidence for synergy between disruption of both *prsA* and *htrA1* leading to major

changes in β-lactam sensitivity. This observation was especially evident in more sensitive agar plate assays (Fig. 2b). The single disruptions of *prsA* or *htrA1* showed comparable minor alteration in sensitivity that was restored by chromosomal complementation (strains *prsA*-C; *htrA1*-C). Disruption of VraSR, which positively regulates both *prsA* and *htrA1* in response to cell wall stress (Supplementary Fig. 6; Supplementary Table 1), or a phosphotransfer signal uncoupling mutation (H156A) of the VraS sensor kinase[33], was noticeably less severe when compared with the double *prsA/htrA1* mutations suggesting that basal expression levels of *prsA* and *htrA1* are important (Table 1). Although COL is sensitive to the latest generation cephalosporin ceftaroline[34], we observed that the double *prsA/ htrA1* mutation also rendered COL hypersensitive to ceftaroline

**Table 1 Microdilution MIC assay with oxacillin**

| COL Oxacillin MICs (mg L$^{-1}$) | 30 °C | 35 °C | 37 °C | 40 °C |
|---|---|---|---|---|
| WT | >512 (512 – >512) | 512 | 256 | 128 (64 – 128) |
| ΔprsA | 512 | 256 (256 – 512) | 128 | 16 |
| ΔprsA-C | >512 | >512 | 512 (256 – 512) | 128 (64 – 128) |
| ΔhtrA1 | >512 | 512 | 128 (128 – 256) | 32 (32 – 64) |
| ΔhtrA1-C | 512 | >512 (512 – >512) | 256 | 64 |
| ΔprsA ΔhtrA1 | >512 (512 – >512) | 128 | 16 (16 – 32) | 4 (4 – 8) |
| ΔprsA ΔhtrA1 + pCN47:prsA$^+$,htrA1$^+$ | >512 (512 – >512) | 512 | 128 | 16 (16 – 32) |
| ΔprsA ΔhtrA1 + pCN47:prsA$^+$,htrA1-S255A | >512 (512 – >512) | 512 (256 – 512) | 128 (64 – 128) | 8 (8 – 32) |
| ΔprsA ΔhtrA1 + pCN47:prsA$^+$,htrA1-ΔPDZ | >512 | >512 (512 – >512) | 256 | 64 |
| ΔprsA ΔhtrA1 + pCN47 | 512 | 128 | 32 | 4 (4 – 8) |
| ΔvraSR | 512 | 256 (256 – 512) | 64 (64 – 128) | 32 (16 – 64) |
| vraS H156A | 512 | 256 (128 – 256) | 128 (64 – 128) | 32 (16 – 64) |
| ATCC 29213 | ≤0.5 | ≤0.5 | ≤0.5 | ≤0.5 |

Data are reported as the modal value with range in parenthesis for at least three independent biological determinations. All concentrations are reported as mg L$^{-1}$

by >4–5 log$_{10}$ using a quantitative colony forming agar plate assay (Fig. 2b).

Plasmid pCN47 alone, or derivatives harboring an engineered prsA-htrA1 dual operon under the control of the pHU house-keeping promoter were used for genetic complementation of the double prsA/htrA1 mutant strain (Table 1; Fig. 2b, c). β-lactam resistance was partially restored using the dual operon plasmid, but not vector alone. Since full complementation of the single mutants was achieved by allelic exchange (Fig. 2b, upper panel) the observed partial complementation in the double mutant strain was likely the result of the plasmid system used and a housekeeping promoter not subject to induction by cell wall stress. Plasmids with mutations in the HtrA1 protease catalytic site (S255A) failed to complement. A plasmid with C-terminal deletion encompassing the HtrA1 PDZ domain showed robust complementation of the double mutant strain, most likely because PDZ domain disruption is known to negatively regulate the protease and so its deletion results in HtrA protease activation[35]. Western blot analysis confirmed the expression of all proteins (Supplementary Fig. 7).

To exclude the possibility that mecA transcription was altered in the mutant strains, we performed quantitative RT-PCR analysis using RNA prepared from wild-type COL, or its mutant derivatives. Our results showed no significant change in mecA exponential growth phase transcription (Supplementary Fig. 8; Supplementary Table 1) indicating that the observed alteration in oxacillin resistance sensitivity was post-transcriptional and most likely post-translational.

When we next examined the oxacillin susceptibility of COL and its mutant derivatives at different temperatures we observed a striking change in drug sensitivity with the double prsA/htrA1 mutation at 40 °C compared to lower temperatures, or each single mutation (Table 1). For example, when comparing COL wild-type with the prsA/htrA1 mutant the recorded MIC difference was identical at 30 °C, whereas we observed a dramatic MIC decrease of 4-fold, 16-fold, and 32-fold over the narrow range of 35, 37, and 40 °C, respectively. This remarkable MIC transition initiates beginning at the same temperatures where we observed onset of PBP2A unfolding by differential scanning fluorimetry. Importantly, none of our strains displayed any temperature sensitivity (tested up to 43 °C) in the absence of oxacillin (Supplementary Fig. 9). These findings are consistent with a model whereby PBP2a folds properly below 35 °C in the presence or absence of extracellular protein folding factors, but requires folding assistance with increasing temperature. It is noteworthy that standard clinical susceptibility testing recommends conditions (microdilution, ≤35 °C, and supplemented with 2% NaCl)

designed to optimize detection of MRSA by resistance assay without any underlying theoretical explanation (www.eucast.org). Collectively, our findings strongly suggest that PBP2a folding is compromised at higher temperatures, or in the absence of extracellular folding factors, or both, thus providing a framework to explain why MRSA detection is enhanced at temperatures ≤35 °C or ionic conditions favoring protein folding.

**HtrA1 is required to remove misfolded PBP2a**. We investigated what specific roles PrsA and HtrA1 performed in the post-translocational maturation of PBP2a. Whole cell total protein extracts, or membrane extracts, were prepared in mid-log phase growth, and normalized to OD$_{600}$. Aliquots were resolved on SDS gels and PBP2a monitored by Western blot analysis. The results are shown in Fig. 2d (uncut membranes are shown in Supplementary Fig. 10 and Supplementary Fig. 5C).

We observed that all COL strains with disruption of htrA1 showed the accumulation of PBP2a degradation fragments as well as a minor fraction migrating in higher molecular complexes, most likely PBP2a-associated peptidoglycan fragments. Disruption of prsA alone did not show degradation fragments demonstrating that these fragments arose specifically in response to disruption of htrA1. Genetic complementation of htrA1 disruption by allelic exchange resulted in the disappearance of degradation fragments. Complementation with the plasmid pCN47-prsA/htrA1-S255A showed degradation fragments indicating that HtrA1 proteolytic activity was necessary for PBP2a quality control, whereas complementation with pCN47 prsA/htrA-ΔPDZ showed no degradation fragments, consistent with experiments showing that removal of this domain activates the HtrA1 protease constitutively[35] (Fig. 2d). Control western blot analysis confirmed expression of all HtrA1 variants (Supplementary Fig. 7).

Anti-PBP2a western blots performed with extracts from wild-type or ΔprsA/ΔhtrA1 mutant cells grown at different temperatures revealed three tendancies consistent with PBP2a misfolding and protein quality control: a reduction of PBP2a in wild-type cells most likely due to clearance of misfolded PBP2a; an increase in full-length PBP2a in the double mutant strain indicative of over-accumulation of misfolded PBP2a that should have been removed; and finally, an increase in PBP2a degradation fragments with temperature (Fig. 2e).

Collectively, we conclude that htrA1 plays a role in PBP2a quality control by directly participating in the removal of misfolded PBP2a. The cleavage patterns observed suggested that other as yet undefined proteases also contribute.

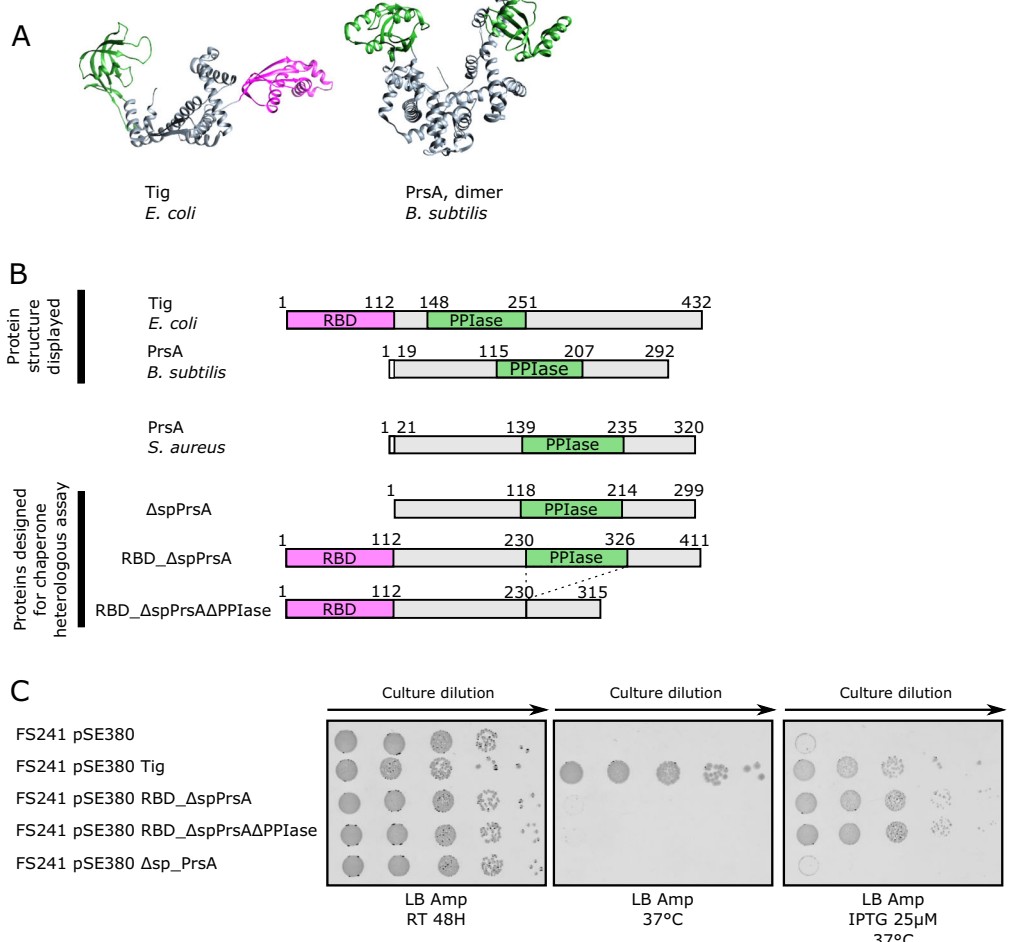

**Fig. 3 a** Ribbon diagram of *E. coli* trigger factor (TF) and *B. subtilis* PrsA. The PPIase domain is shaded *green*, the ribosome binding domain (RBD) of *E. coli* trigger factor *magenta*. Both PrsA and TF are reported as biological dimers[36,40]. Protein database accession 2MLX (TF) and 4WO7 (PrsA). **b** Schematic domain representation of TF and PrsA shown in *panel A*, together with the TF/PrsA chimeric hybrids constructed. Δsp denotes deletion of *S. aureus* PrsA signal peptide. **c** Colony forming viability assay on agar plates and serial 10× fold dilution in various conditions. Strain FS241 is *E. coli* MC4100 Δ*dnaK*, Δ*dnaJ*, Δ*tig*[41] and cannot grow above 25 °C without a source of functional trigger-factor chaperone activity. The IPTG-inducible vector pSE380, alone, or with the indicated inserts as in panel **b** were introduced into FS241 at the permissive temperature then tested for growth in restrictive conditions (37 °C) with or without IPTG (25 μM)

**PrsA has trigger factor-like anti-aggregation activity.** We next asked what specific role PrsA played in PBP2a maturation by testing whether PrsA possessed detectable chaperone activity, apart from its known parvulin-like PPIase domain[36]. This was accomplished by designing a heterologous functional genetic assay in *E. coli*. Trigger factor (TF) is an anti-aggregation chaperone PPIase tethered to ribosomal protein L23 and that engages nascent polypeptides as they emerge from the ribosome[37,38]. Structural analysis showed that TF transiently interacts with nascent polypeptides using a folding platform on which extended polypeptides interact reversibly with hydrophobic patches[39,40]. A hypersensitive *E. coli* strain, FS241, lacking *dnaK* (Hsp70), *dnaJ* (Hsp40), and *tig* (TF) does not grow at temperatures above 25 °C[41]. Structure/function study of TF with this strain showed that growth at 37 °C could be restored by wild-type TF, or TF lacking its PPIase domain, indicating that the major contribution from TF comes from provision of an anti-aggregation folding platform function and not apparently its associated PPIase activity[41]

Despite the absence of overall amino acid sequence similarity, PrsA from *B. subtilis* and *E. coli* TF share three-dimensional structural resemblance with a central folding cavity platform, and

since the PPIase domain occupies one tip of the molecule[36] (Fig. 3a). The TF ribosome binding domain (RBD) is positioned at the opposite extremity. Nascent polypeptide chains predictably encounter similar folding problems when exiting from the ribosome or from the general secretory pathway translocation channel. We hypothesized that if PrsA were to possess TF-like properties as an anti-aggregation chaperone, then an engineered *S. aureus* PrsA lacking its lipoprotein signal sequence should functionally substitute in *E. coli* by replacing TF for nascent chain polypeptide protection on the ribosome.

We constructed plasmids expressing *S. aureus* PrsA, or PrsA fused in frame to *E. coli* trigger factor RBD. Plasmids were transformed into FS241 at 23 °C, then subsequently tested for growth in colony forming assay at 37 °C with, or without, IPTG inducer (Fig. 3b, c). We observed that *prsA*, or *prsA* lacking its PPIase parvulin domain when fused in frame to the *E. coli* TF RBD could fully complement strain FS241 for growth at 37 °C. We could not find conditions where *prsA* by itself, without RBD fusion, could restore growth of cells at 37 °C. Control vector alone could not complement, whereas a control plasmid containing wild-type *E. coli* *tig* could complement, even in the absence of inducer. We conclude from this analysis that PrsA possesses

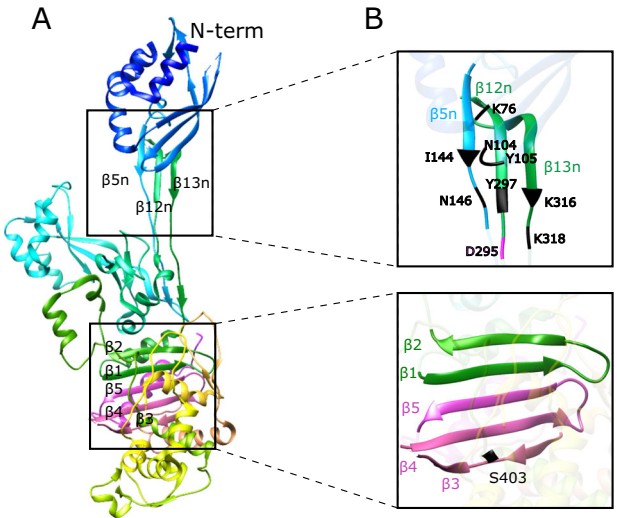

**Fig. 4 a** Ribbon diagram of PBP2a using spectral shading from the N-terminus membrane proximal domain (top, blue) to the transpeptidase catalytic domain (bottom, magenta), PDB 1MWR. Boxed regions at left highlight the two anti-parallel β-sheet regions situated in the allosteric domain (β5n-β12n-β13n) and the catalytic transpeptidase domain (β1-2, β3-5) each composed of particularly discontinuous polypeptide segments. **b** The right upper panel depicts details of the folding architecture of a portion of the PBP2a allosteric site indicating amino acids found important for contact with ceftaroline, quinoxalinone, and D-ala-D-ala[13,43]. D295 and the base of β12 are shown and are thought to initiate the allosteric trigger by molecular dynamics simulation[44]. The bottom panel shows the active site serine 403 in the catalytic domain as well as the five strand discontinuous anti-parallel beta strand fold

robust chaperone anti-aggregation activity comparable to *E. coli* TF, and furthermore, as shown with TF structure function analysis[41], our detected PrsA chaperone activity was independent of the presence of its parvulin-like PPIase domain. Biochemical analysis of purified PrsA from *B. subtilis* has so far failed to detect anti-aggregation activity in vitro[36]; our in vivo analysis now demonstrates that PrsA indeed must possess such activity.

**Tertiary folds of PBP2a suggest a need for chaperones**. We considered whether there are any prominent structural features of PBP2a that suggest a need for chaperones. Inspection of the PBP2a three-dimensional structure[42] revealed at least two possibilities involving anti-parallel β-sheets composed of particularly discontinuous polypeptide segments (Fig. 4). First, the PBP2a N-terminal domain connects to the remaining molecule by a three-strand β-sheet (β5n-β12n-β13n) that must align in register and pack with N-terminal domain α-helix 4 and β-1-3 to form the allosteric trigger (Fig. 4). This three-strand β sheet binds ceftaroline, quinazolinone2A, and D-ala-D-ala Lipid II stem[13,42,43]. A span of 149 amino acids separates the β5n sequence (140–147) from its β12n partner strand (296–303). Thus, during the secretion process β5n must necessarily await the emergence of its β12n/13n partner strands underscoring a potential critical point where PrsA-mediated anti-aggregation could intervene. Molecular dynamics simulation shows that the allosteric trigger likely initiates near D295 at the base of β12n so the configuration and alignment of this β-sheet structure is critical to the overall function of the enzyme[44]. Secondly, another discontinuous five strand anti-parallel β-strand fold occurs in the core catalytic domain of PBP2a: β1-2 (345–363) β3-5 (594–634), and by extension, likely all PBP catalytic domains and PASTA domains that share this conserved fold configuration.

## Discussion

Collectively, our results indicate that a functionally active PBP2a can generate a folding conformer that results in an irreversible insoluble inactive entity under experimental temperature and pH conditions in vitro. The existence of this catastrophic off-pathway conformer arising from a fully folded functional PBP2a following minor temperature shift and/or pH explains the critical requirement for a quality control system that must prevent this if *S. aureus* MRSA is to remain β-lactam resistant. Thus, PrsA and HtrA1 together contribute to the proper maturation of PBP2a during its vectorial transit from the secretion translocation channel. We cannot, of course, exclude that other factors contribute to PBP2a maturation and await discovery. It is worthwhile remarking that the interval of 37–40 °C corresponds to fever onset conditions, or elevated basal body temperatures of certain animals (i.e., rabbits) used for experimental infection models. It is conceivable that PBP2a biological activity is partially compromised in these situations.

The need for allostery that underlies the mechanism of methicillin resistance[11] imposes a cost on PBP2a that brings its ability to bring resistance to β-lactams dangerously close to its intrinsic unfolding and consequent inactivation threshold. The N-terminal allosteric domain is particularly sensitive to proteolysis, and the importance of this region, especially mutation of motif 3 immediately preceding the β12n sheet for protein activity and folding was highlighted in study of other class B PBPs in other organisms[45].

We propose a model highlighting a role for at least two extracellular folding factors governing the acquisition of the MRSA phenotype. On the outer face of the membrane PrsA lipoprotein provides a transient anti-aggregation platform for the PBP2a polypeptide as it vectorially exits the SecYEG translocon channel. When *prsA* alone is disrupted, a fraction of PBP2a can still fold spontaneously, and misfolded variants removed by HtrA1. This would explain the minor effect on MIC observed with Δ*prsA*. Alternatively, in Δ*htrA1* strains, PrsA can still assure an anti-aggregation folding platform, and apparently accumulated misfolded polypeptides do not markedly disrupt function, again explaining the minor effect of *htrA1* disruption alone on MIC. Loss of both PrsA and HtrA1, however, severely compromises the proper folding of PBP2a and results in a level of enzyme, or enzyme conformers, unable to function adequately in the presence of β-lactam antibiotics. We cannot exclude at this time that *S. aureus* HtrA1 also contributes to PBP2a folding as a chaperone itself, or that other folding factors are involved. Minor increase in temperature further exacerbates misfolding of PBP2a and shifts oxacillin MIC in the direction of increased sensitivity. The temperature effect is magnified by either single mutant and drastically impacts the β-lactam resistance phenotype of the *prsA/htrA1* double mutant.

Apart from restoring sensitivity to oxacillin, dual disruption of *prsA/htrA1* in our study also resulted in enhanced hypersensitivity to ceftaroline, an anti-MRSA cephalosporin antibiotic first commercialized in 2010. Colony viability was remarkably reduced by several orders of magnitude. Clearly, modulating PBP2a post-translocational folding can not only dramatically restore sensitivity to β-lactams, but also possibly contribute to enhanced therapeutic efficacy by compromising the emergence of resistance.

Our model predicts that HtrA1 recognizes features of PBP2a that underlies the critical substrate discrimination and quality control process, most likely the exposure of normally buried hydrophobic amino acid residues. Recent work showed that PBP2a is associated with membrane microdomains and that organization was affected by disruption of flotillin and pigment lipid synthesis[46]. Thus, the native mature functional form of

PBP2a may also require quaternary interactions, and perhaps association with other protein partners to organize properly the peptidoglycan biosynthetic apparatus. It is worthwhile noting that *Streptococcus pneumoniae* PBP2x displays allostery[47]. Thus, the dependence of the critical folding of important Class B family PBPs necessary for β-lactam resistance upon extracellular chaperone quality control may extend to a larger group of pathogenic organisms.

## Methods

**Bacterial strains, biosafety, and culture conditions**. All strains used in this study are listed in Supplementary Table 2. *Escherichia coli* strains were grown in Luria-Bertani broth (LB), and *Staphylococcus aureus* strains were grown in Mueller-Hinton broth or agar (cation-adjusted, Difco). All biosafety level 2 work with *S. aureus* was conducted under the auspices of the Swiss Federal Institute of the Environment regulations and guidelines and formal authorization attributed to the PI (WLK). All laboratory work was conducted in confined conditions, laminar flow biosafety cabinets, and with appropriate personal protection equipment. Bacteria and surfaces were inactivated with Deconex 50 FF (Borer Chimie, Swizerland, 4% aqueous final concentration) or BioSanitizer S (SaniSwiss, Geneva, Switzerland). Biowaste was autoclaved within the building prior to off site incineration conducted by our medical center waste management logistics team. Growth media were supplemented as appropriate with ampicillin (100 μg mL$^{-1}$), kanamycin (50 μg mL$^{-1}$), erythromycin (3–5 μg mL$^{-1}$), spectinomycin (100–140 μg mL$^{-1}$), lincomycin (20 μg mL$^{-1}$), or chloramphenicol (15 μg mL$^1$). Strain constructions were performed by generalized transduction using bacteriophage Φ11, Φ80α, and Φ85. Plasmids prepared in *E. coli* K-12 were introduced into *S. aureus* strains by electroporation first into either non-restricting RN4220, or transformation into *E. coli* IM08B[48]. Routine recombinant DNA work used *E. coli* DH5α for all plasmid constructions. Strain verifications were performed with appropriate PCR primers and colony PCR or purified gDNA (Supplementary Table 3) together with automated DNA sequencing. Antibiotics were obtained from Sigma, the Geneva University Hospital pharmacy, Oxoid, or prepared from ceftaroline fosfamil (Zinforo) by alkaline phosphatase treatment of the prodrug and differential precipitation as described[10].

**PBP2a purification**. The *mecA* sequence encoding PBP2a, but lacking its transmembrane anchor sequence (Δ1–22; thus 23–668), was amplified by PCR from COL gDNA using primers incorporating Nco and HindIII restriction sites (Supplementary Table 3) to clone in the vector pET24d (Novagen). The expression vector provided in-frame fusion to a C-terminal 6xHis tag and was fully sequence verified. Recombinant proteins were prepared from *E. coli* strain Tuner (λDE3) (Novagen) grown in 1 L LB supplemented with 0.2% glucose followed by mid-log phase induction with 1 mM IPTG and 18 h growth at 30 °C. Cells were harvested and resuspended in 10 ml lysis buffer (25 mM Tris HCl pH 7.5, 50 mM NaCl, 1 mM PMSF, 5 mM β-mercaptoethanol) per liter of cells and lysed with hen egg white lysozyme (200 μg mL$^{-1}$) and two cycles of freeze-thawing at −80 °C. All subsequent steps were performed at 4 °C. Thawed extracts were sonicated briefly on ice to reduce viscosity, then clarified soluble extracts prepared by ultracentrifugation in a Beckman Type 70Ti rotor 100,000 × g for 30 min. Supernatants were collected and precipitated by the addition of solid ammonium sulfate to 50% saturation (29.5 g 100 mL$^{-1}$). Pellets were collected by low speed centrifugation and resuspended with Buffer A (50 mM sodium phosphate pH 7.4, 150 mM NaCl) supplemented with 10 mM imidazole and PBP2a purified using ThermoFisher Ni-NTA HiTrap 6% agarose beads. PBP2a was eluted with 0.5 M imidazole. The eluate was subjected to sequential precipitations of 30, 40 and 50% saturation ammonium sulfate. PBP2a remained soluble in 50% ammonium sulfate and was essentially pure. PBP2a was extensively dialyzed to remove residual imidazole in Buffer A and then concentrated by application to a 3 ml Macro-Prep High S Support (Bio-Rad) column equilibrated in Buffer A. PBP2a was step eluted in 0.4 M NaCl, dialyzed in Buffer A and concentrated in a Millipore Amicon Ultra 10 K MWCO centrifugal cell. PBP2a concentrations were determined by Bradford assay and using the molar extinction coefficient for the TM-domain truncated protein: 81290 M$^{-1}$ cm$^{-1}$ at 280 nm[49].

**PBP2a bocillin FL derivatization and allosteric competition**. Purified PBP2a was covalently derivatized on its active site serine by exposure to the fluorescent penicillin V analog, bocillin-FL[23]. Briefly, PBP2a at 2 μM in a 20 μL reaction volume in Buffer A was pre-incubated at the indicated temperature for 15 min and then 1 uL of bocillin FL stock at 1.5 mM was added (71 μM final concentration) and incubated an additional 5 min. Reactions were stopped by the addition of an equal volume of cold 15% (v/v) trichloroacetic acid and pellets harvested in an Eppendorf microcentrifuge. Pellets were washed in ice cold 70% ethanol, and dried. Samples were resuspended in SDS sample buffer and resolved on 12% SDS polyacrylamide gels in the dark. Gels were soaked in deionized water for 15 min and then scanned in a Bio-Rad Chemidoc MP imaging system using Alexa 488 preset detection conditions. Gels were then stained with Coomassie Brilliant Blue. For competition assays, 1 or 10 μL ceftaroline or oxacillin (stock 1.5 mg mL$^{-1}$ and

1.0 mg mL$^{-1}$, respectively) was added for 5 min prior to the addition of bocillin FL. Stock concentrations of both drugs were adjusted so molar concentrations were equivalent. Final reaction conditions were: PBP2a at 2 μM; bocillin FL at 71 μM (35× excess). Ceftaroline or oxacillin were used at 188 μM, or 826 μM final concentrations in the competition assays.

**PBP2a thermal aggregation assay**. Phosphate buffer stocks at pre-determined pH in the range 7.4–5.8 were prepared by mixing of stock 1 M solutions of monobasic and diabasic sodium phosphate. PBP2a at 2 μM final concentration was prepared in each buffer condition (prepared at 50 mM sodium phosphate 150 mM NaCl). Samples (30 μL) were incubated for 15 min at the indicated temperature in a digitally thermostated dry block. Samples were separated into pellet and supernatant fractions by centrifugation in an Eppendorf microfuge (10,000 × g, 3 min). The supernatant (soluble) fraction was precipitated by addition of an equal volume of 15% trichloroacetic acid, centrifuged, washed in 70% ethanol and resuspended in SDS sample buffer. The pellet (insoluble) fraction was washed once with reaction buffer pre-heated to the indicated temperature. Pellet fractions were resuspended in SDS sample buffer. All reactions were resolved on 12% SDS PAGE gels followed by staining with Coomassie Brilliant Blue. For the aggregated PBP2a pellet activity assay insoluble pellets prepared as above were collected by centrifugation, washed with pre-heated reaction buffer to remove residual supernatant, and then subjected to bocillin FL derivatization and allosteric competition assay as above to determine their activity.

**Differential scanning fluorimetry**. Purified PBP2a was diluted in 50 mM sodium phosphate buffer, 150 mM NaCl at the indicated pH. Samples (45 μL) of protein at 5 μM concentration were mixed with 5 μL of 100X Sypro Orange (Sigma–Aldrich, 5000×, ref S5692). Thermal fluorimetry scanning was performed in 96-well plates (Bio-rad HSP901 0.2 ml) on a CFX96 real-time PCR detection system (Bio-rad). The temperature was ramped from 20 to 95 °C in 0.5 °C increments (15 s each). Data were analyzed with the Bio-Rad CFX Manager v3.1 and GraphPad Prism v8. The results are reported as triplicate independent determinations with baseline subtractions of signal from controls lacking protein. The increase in fluorescence was displayed as a first derivative plot as a function of temperature and the inflection point taken as the melting temperature.

**Partial proteolysis assay**. PBP2a samples (30 μL) were prepared at 2 μM in Buffer A, pre-incubated at the indicated temperature for 15 min, and then subjected to mild proteolytic digestion with the addition of 0.6 μg trypsin or proteinase K (representing a protease:PBP2a ratio of 1:50 by mass in the reaction volume). Aliquots (20 μL) were removed at the indicated time points and stopped by the addition of an equal volume of ice cold 15% trichloroacetic acid. Samples were precipitated by centrifugation, washed in 70% EtOH, dried and resuspended in SDS sample buffer. Proteins were resolved on 12% SDS gels. The detection of stable domains was monitored by limited proteolysis of PBP2a previously derivatized with bocillin FL followed by Western analysis using either anti-PBP2a monoclonal antibody, or monoclonal anti-6xHis-HRP conjugate antibody. The reaction was scaled 5-fold for this experiment, and then samples processed by trichloroacetic acid precipitation as above.

**Antibiotic susceptibility tests**. Broth microdilution MICs were performed according to EUCAST (European Commission on Antibiotic Susceptibility Testing) guidelines in a 96-well microplate in cation-adjusted Mueller-Hinton II broth (CAMHB, Becton Dickinson, Difco). Briefly, a 0.5 McFarland standard cell suspension was prepared from a 24 h agar culture in NaCl 0.9% using a bioMérieux Densimat apparatus (bioMérieux, France). After 1:100 dilution in CAMHB, 50 μL was added to 50 μL of 2X oxacillin solution to obtain a final concentration range from 0.5 to 512 mg L$^{-1}$. Microplates were incubated for 24 h at either 30, 35, 37, or 40 °C. *S. aureus* MSSA (methicillin sensitive) strain ATCC 29213 was used a quality control. Determinations were performed in triplicate assay and the composite data reported as the modal value together with the range from a minimum of three independent biological determinations.

**Colony formation counting**. CFU counts were performed by spotting aliquots (5–10 μL) of 10-fold serial dilutions onto agar plates and incubated at the indicated temperature and colony counting after the indicated time. Serial 10-fold dilutions of *S. aureus* overnight cultures adjusted to McFarland standard 1.0 (3.0 × 10$^8$ cfu mL$^{-1}$) were made using a turbidity Densimat apparatus (bioMérieux, France) and sterile saline. *E. coli* overnight cultures were OD$_{600}$ normalized with sterile LB prior to serial dilution.

**Total protein and membrane extractions**. Overnight cultures of strains were diluted 1:100 in 20 mL fresh cation-adjusted Muller-Hinton broth (Difco) in 50 ml Falcon tubes and grown at 37 °C (or at indicated temperature) with vigorous aeration to OD$_{600}$ 1.0. The OD$_{600}$ for each culture was recorded, cells harvested for 10 min at 5000 rpm, and pellets washed once in sterile phosphate buffered saline (PBS). After re-centrifugation, the pellets were frozen at −80 °C. For cell lysis, pellets were thawed and resuspended in 20 μL of lysis buffer per OD unit (Lysis

buffer: PBS with complete EDTA-free protease inhibitor cocktail (Roche), DNase I (Invitrogen) to 200 µg mL$^{-1}$ final concentration and lysostaphin (AMBI products) to 200 µg mL$^{-1}$ final concentration). Tubes were incubated at 30 °C for 1 h. Samples were transferred to 1.5 ml Diagenode microtubes and sonicated in a Bioruptor device for 20 cycles of a 30 s sonication then 30 s rest program with sample tube immersion recirculating refrigeration at 4 °C. Unlysed cells were removed by low speed centrifugation in an Eppendorf refrigerated centrifuge (5000 rpm for 10 min). The supernatants were designated as total extract and protein content quantified by Bradford Assay (Bio-Rad) using bovine serum albumin as a standard. An aliquot (300 µL) of the total extract was further fractionated by ultracentrifugation in a Sorvall RC M150 centrifuge in a Beckman S45A rotor at 40,000 rpm 1 h (100,000 × $g$). The supernatant was carefully removed and the membrane fraction was obtained by pellet extraction using 150 µL extraction buffer (25 mM Tris HCl pH 7.5, 150 mM NaCl, 1 mM MgCl2, 30% glycerol, 1% (v/v) Triton X-100, 0.5% (v/v) sodium N-lauryl sarcosine) and incubation for 12 h at 4 °C and sonicated. Protein content was quantified by Bradford assay.

**Immunoblotting.** Aliquots of cell-free extracts (15 µg of membrane protein extract for PBP2a or 10 µg of total protein for PrsA and HtrA1) were resolved in 10% SDS PAGE gels and transferred to Porablot NCP nitrocellulose membranes (0.45 µm; Machery-Nagel, Switzerland) by semi-dry transfer (Bio-Rad). Membranes were stained with Ponceau Red and photographed to assure transfer uniformity. Membranes were blocked with non-fat milk 5% w/v in PBS-0.2 % Tween-20 washed in PBS/Tween. Proteins were detected using respectively: mouse mono-clonal anti-PBP2a (bioMérieux France, Slidex MRSA detection kit, ref 73117) (dilution 1:500), rabbit anti-PrsA[20] (dilution 1:50,000) or rabbit anti-HtrA1 (Eurogentec, Belgium, custom anti-peptide antibody: KLH-conjugated C + TNNKGGNQLDGQSKK) dilution 1:20,000). Secondary antibodies, HRP-conjugated goat anti-mouse or anti-rabbit Ig (Bio-Rad) were diluted 1:5000 in PBS-Tween. His-tag purified PBP2a was detected using HRP-conjugated mouse monoclonal anti-HIS antibody (Invitrogen). HRP-conjugated mouse monoclonal anti-HIS antibody (Invitrogen) was used at 1:500. Blots were washed and developed using West Pico enhanced chemiluminesence according to the manufacturer's recommendations (Pierce).

**Construction of PrsA-HtrA1 dual gene expression vector.** A synthetic dual operon of prsA with either htrA1 WT or htrA1 S255A active site mutant or truncated htrA1 lacking its PDZ domain were generated by overlap PCR using appropriate primers (Supplementary Table 3). All htrA1 coding sequences used htrA1 Kpn OUT template lacking the internal KpnI restriction site prepared by overlap PCR. PCR products were cloned in Bluescript pKS(+) for sequence verification and then put under the control of the housekeeping nucleoid HU promotor by subcloning (KpnI, PstI) into pMB-HU vector[50,51]. Cassettes were excised and subcloned into the final pCN47 plasmid[52] digested by PstI and partially with NotI. Expression plasmids were electroporated into RN4220 and subsequently transduced using Φ80α into the COL ΔprsAΔhtrA1 clone.

**E. coli trigger factor (TIG)-S. aureus PrsA fusions.** Polymerase chain reaction (PCR) was performed with gDNA from S. aureus strain 8325-4 (AR1089[53]) with primers as indicated in Supplementary Table 3 in order to amplify S. aureus prsA lacking its lipoprotein signal sequence (Δ1–21) with the addition of an in-frame N-terminal methionine codon and N-terminal BamH1 and C-terminal Xba1 restriction sites. The same primers were used to amplify prsA coding sequence from AJ843 gDNA lacking the parvulin PPIase domain coding sequence[22]. Fragments were gel purified, digested with BamH1 and XbaI (forward primer contained BamH1 site, reverse primer contained an XbaI site) and then cloned into the IPTG-inducible E. coli expression plasmid pSE380ΔNco1 (Invitrogen)[54]. Expression plasmids with S. aureus prsA fused in-frame C-terminally with the native E. coli trigger factor (tig) ribosome binding domain (tig RBD, amino acids 1–148) were constructed in a two-step sowing PCR using the primers as indicated in Supplementary Table 3. The forward primer in this case contained an EcoR1 site consistent with the original cloning of pSE380ΔNco-tig[41,54]. The fragments were gel purified, digested with EcoR1 and XbaI and cloned into pSE380ΔNco cleaved with the same enzymes. All constructs were sequence verified with appropriate primers. Recombinant plasmids, including empty pSE380ΔNco vector and a positive control pSE380ΔNco containing wild-type E. coli tig[41,54] were transformed into E. coli strain FS241 without heat shock and colonies obtained by incubation at the permissive temperature 22 °C for several days[41,54]. Single colonies were picked and grown in LB broth again at permissive temperature for several days. Spot test viability assays were performed using OD$_{600}$ normalized cultures that were serially diluted 10-fold in fresh LB broth. Aliquots (10 µL) were spotted on LB agar plates containing ampicillin (100 µg mL$^{-1}$) with or without IPTG and incubated at various temperatures. Pilot experiments were run to determine optimal IPTG concentrations for all genetic complementations.

**kanA marker insertion nearby htrA1.** To facilitate genetic complementation of htrA1::cam by bacteriophage-mediated allelic exchange, a site-specific insertion of kanA was engineered in the SAOUHSC_01833-SAOUHSC_01835 intergenic

region at coordinate 1739594 (sequence reference NC_007795 NCBI Genbank) 64 bp downstream of the serA (SAOUHSC-01833) stop codon and 3.7 kb upstream of the htrA1 (SAOUHSC_01838) start codon. A unique BamH1 restriction site was engineered in the pBT2[55] targeting vector at this site by PCR using appropriate flanking regions using primers in Supplementary Table 3. The detailed methods of the site-specific chromosomal insertion of a kanamycin resistance cassette encoding kanA have been described[51]. HtrA1 allelic exchange was performed by generalized bacteriophage-mediated transduction using bacteriophage Φ80α. Candidate transductants were verified by PCR assay of the restored wild-type htrA1 allele.

**RNA extraction.** Culture aliquots (1 ml) were collected at OD$_{600}$ 0.5, centrifuged for 3 min at 8000 rpm and the pellet was washed once with 1 mL 1X TE buffer (10 mM Tris HCl pH 8.0, 1 mM EDTA). The pellet was resuspended in 100 µL lysostaphin buffer composed of 200 µg mL$^{-1}$ lysostaphin, 200 µg mL$^{-1}$ DNase I, 40 U RNasin Plus Ribonuclease inhibitor (Promega) in 1X TE and placed at 37 °C for 10 min. RNA was then extracted using RNeasy Plus Mini Kit (Qiagen) following the manufacturer's instructions and adding two modifications: cell homogenization was performed with QIAshredder (Qiagen) columns after adding RTL Plus lysis buffer and DNase I treatment using RNase-Free DNase Set (Qiagen) was applied after first column wash.

**Quantitative reverse transcriptase PCR (qRT-PCR).** Absence of DNA contamination in total RNA extractions was always assessed by qPCR using 2X Takyon for probe assay (Eurogentec, Belgium). Primers and probes were designed using PrimerExpress software (Applied Biosystems) and obtained from Eurogentec (Supplementary Table 3). Total RNA (4 ng) per well and a final concentration of primers of probes of 0.2 µM were used to obtain an efficiency of 100% (amplification factor of 2). Measurements of mRNAs were determined by quantitative reverse transcriptase PCR (qRT-PCR) using the Platinum qRT-PCR ThermoScript One-Step system (Invitrogen, Carlsbad, CA). The mRNA levels were normalized to 16 S rRNA levels, which were assayed in each round of qRT-PCR as internal controls. The statistical significance of strain-specific differences in normalized cycle threshold ($C_T$) values of each transcript was evaluated by Student paired $t$-test, and data were considered significant when $P$ was <0.05. For convenience in figure presentation, data were plotted to reflect fold change and error bars correspond to ±SEM for at least three independent determinations of cycle thresholds. Nevertheless, the reported $P$-values always correspond to calculations with the normalized cycle thresholds.

**Statistics and reproducibility.** For MIC assays, modal values are reported together with the range in parenthesis. The data represents three independent biological determinations. Bacteria used for culture were obtained in each instance from single colony isolation from −80 °C master stocks. qRT-PCR assays were performed with three independent biological determinations representing single colony isolation, culture growth, and RNA preparation. The data in Supplementary Fig. 6 used Student's paired $t$-test for $n = 3$ determinations, but with the Bonferroni correction applied. The data in Supplementary Fig. 8 used Student's paired $t$-test for $n = 3$ determinations. For the results reported herein for all experiments, no data were excluded. MRSA strains cultivated in the laboratory are known to have unstable SCCmec elements encoding mecA. Strain integrity was therefore verified by PCR-based assay for mecA sequence and ccr for the SCCmec element using primers as described in Supplementary Table 3. Although pCN47-based plasmids harboring htrA1 could be constructed, they could not be transformed into the appropriate S. aureus mutant strains for unknown reasons. pCN47-based plasmids were therefore constructed for dual expression of both prsA and htrA1 in a hybrid operon format to expedite complementation analysis of the double mutant strains.

**Graphics.** Representations of Protein database (PDB) structure images were made with the Chimera graphics software package v 1.12–1.15[56] (University California San Francisco). Protein data base (PDB) accession 1MWR for PBP2a; PrsA 4WO7, and trigger factor 2MLX.

**Reporting summary.** Further information on research design is available in the Nature Research Reporting Summary linked to this article.

## Data availability
All data used in this study are included within the main text or provided in the supplementary files. Any additional information and biological material described in the study is available from the corresponding author upon request. The Supplementary Information file contains the underlying data for qRT-PCR analysis as well as uncut gel images. The source data for differential scanning fluorimetry reported in Fig. 1d are provided in an Excel file format in Supplementary Data 1.

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

## Acknowledgements

This work was supported by funds from the Swiss National Science Foundation 310030-146540, and 310030-166611 to W.L.K., 310030-169404 to A.R., a Novartis Consumer Health Foundation postdoctoral fellowship grant to E.L., The Ernst and Lucie Schmidheiny Foundation to W.L.K., and the Canton of Geneva. We thank Alexandra Gruss (INRA U. Paris-Saclay) and Françoise Schweiger (U. Geneva) for bacterial strains, Vesa Kontinen (U. Helsinki) for the gift of anti-PrsA antibody, and Pierre Genevaux (CNRS Toulouse) for valuable advice. We gratefully acknowledge the contribution of Elodie Durand to experiments herein during her semester practical training stage in our laboratory under the auspices of the Biotechnology Technical Program (BTS) Lycée Albert Camus, Nîmes, France.

## Author contributions

M.R. performed experiments, analyzed data and helped prepare the manuscript. E.L.-A. performed experiments and analyzed data. O.O.P. performed experiments and analyzed data, and helped in experimental design. R.S. performed experiments, analyzed data, and helped prepare the manuscript. A.R. performed experiments, helped in the experimental design, and helped prepare the manuscript. W.L.K. conceived the project, wrote the manuscript, analyzed data, and performed experiments. W.L.K. provided overall funding, project supervision, and administration.

## Competing interests

The authors declare no competing interests.
