## [Peer Review File · Communications Biology]

Reviewers' comments:

Reviewer #1 (Remarks to the Author):

In their study Roch et al. use complementary genetic, structural and biochemical approaches to show that PBP2A activity is impacted by a catastrophic folding transition, necessitating active interactions with the extracellular folding factors PrsA and HtrA1 to ensure proper post-translocational PBP2A maturation and sustained MRSA β -lactam resistance. The results presented in this study could inspire novel approaches to counteract β -lactam resistance in MRSA, by targeting the extracellular protein folding machinery.

Overall, the experimental evidence presented in this paper convincingly shows that functional PBP2A maturation is reliant on interactions with folding-factors at physiological conditions following its vectorial transit through the secretion translocation channel. While the authors have previously highlighted the role of PrsA in the extracellular maturation of PBP2a (doi:10.1128/aac.06264-11, doi:10.1128/aac.02333-15), the comprehensive characterization of HtrA1 and PrsA involvement in the extracellular PBP2A maturation presented in this study provides novel insights into this important prerequisite for β -lactam resistance in MRSA. There are, however, some minor points that need to be addressed.

1. The introduction of the extracellular folding machinery (HtrA, PrsA) in the introduction (line 57) as well as in the result section (line 129) is abrupt. An introduction of these factors that is more cohesive with the previous paragraphs would help to more efficiently convey the authors experimental rationale.
2. The authors claim that purified PBP2A was stable at 5mg/ml and 4C for up to five months (lines 97-98). This claim should be supported by the corresponding data or a relevant reference.
3. The results presented in Fig4 (VraTSR regulation of htrA1 and prsA) need more context than that provided at first mention (line 135). The authors provide a comprehensive interpretation of their results later (lines 148-152). As the first mention of VraTSR is solely aimed at establishing its regulatory control over htrA1 and prsA reference of one of the many publications providing evidence for this relationship here might be more appropriate (e.g. Boyle-Vavra Antimicrob Agents Chemother 2013 or their own Antimicrob Agent Chemother publication from 2012).
4. References to VraTSR should be harmonized throughout the paper. First, the regulatory system is generally referred to as VraTSR and not VraRST (as in line 135). Second, the authors use the nomenclature vraTSR (line 135) and vraRS (line 148) interchangeably. For consistency nomenclature should be harmonized.
5. The authors claim that coordinate expression under cell wall stress conditions is 'strong evidence' for cooperation (lines 136-137). However, a multitude of genes is co-regulated under cell wall stress conditions, not all of which cooperate. Therefore, coordinate expression alone, specifically in response to an external stressor, is not necessarily evident of cooperation and this statement should be rephrased.
6. The authors claim that disruption of VraTSR 'was noticeably less severe when compared with either single or double prsA/htrA1 mutation' (lines 148-152). However, the data provided in Table 1 do not convincingly support this claim, but show that at 37°C COL Δ vraSR is actually more susceptible to Oxacillin (MIC 64, range 64-128) when compared to COL Δ htrA1 (MIC 128, range 128-256) or COL Δ prsA (MIC 128). Therefore, the authors claim should be rephrased or removed.

7. The authors claim that the dual *htrA1/prsA* operon on plasmid pCN47 does 'restore resistance to β -lactams' (lines 158-159). While their data support restored resistance against low concentrations of Oxacillin (32mg/ml) the complemented double mutant remains substantially more susceptible to CPT and high levels of OXA (128 mg/ml) when compared to wild type or complemented single mutant strains. While this observation could represent an artefact resulting from the housekeeping promoter used to drive *htrA1/prsA* expression, this fact should be acknowledged in the interpretation of the results.

8. The authors should provide a reference for functional/structural studies that have shown restored growth of hypersensitive *E. coli* strains when complemented with trigger factor (lines 224-225).

9. The reference to Fig 4 in line 258 is unnecessary.

10. The statement that '*PrsA* and *HtrA1* together assure the proper maturation of PBP2A during its vectorial transit from the secretion translocation channel', should be rephrased as the authors cannot exclude contribution of additional factors during this process.

11. The authors claim that 'Colony viability was remarkably reduced to a level that would predictably compromise even the survival of rare spontaneous mutants conferring altered resistance to ceftaroline'. Moreover, they state that modulating PBP2A post-transcriptional folding could 'possibly contribute to enhanced therapeutic efficacy by compromising the emergence of resistance'. For both of statements the authors fail to provide any data that would support their claims. Until now there has not been an evolution-proof antimicrobial therapy, but every drug/drug combination has eventually been met with the rise of resistant clones. Therefore, these statements should be rephrased or removed.

12. The legend of Supplementary Figure 2 does not indicate which part of the legend corresponds to subpanel A or B.

13. Supplementary Figure 3B should include the MIC data of the wild type strain for context.

Language suggestions:

Line 127 – Remove 'therefore'.

Line 160 – Remove the double spacing at beginning of sentence.

Line 317 – 'Thus, the dependence of the critical folding of important ... for β -lactam resistance upon extracellular... ' instead of 'Thus, the critical folding ... for β -lactam resistance dependent upon extracellular ... '.

Reviewer #2 (Remarks to the Author):

Roch and colleagues from the Kelley Group further investigated the possibilities of studying chaperone proteins in *S. aureus* MRSA for biochemical frailties in the PBP2A protein.

Their work in this manuscript described several experiments:

- 1) Perturbations of PBP2A 3D structure using physiologically significant modulations of heat and pH.
- 2) Effects on beta-lactam resistance via perturbations of *PrsA* and *HtrA1* proteins and their roles as chaperones in proper folding and anti-aggregation in MRSA COL. Complementation with plasmids for these proteins validated their roles in beta-lactam resistance. The use of Ceftaroline (latest-gen cephalosporin with anti-PBP2A activity) and Oxacillin (earlier non-anti-PBP2A penicillin) were appropriate test agents for beta-lactam drugs in MRSA.

- 3) Use of Western blot analysis to confirm aggregation of misfolded PBP2A without functional HtrA1.
- 4) PrsA protein appears to have homologous function in *S. aureus* to Trigger factor in *E. coli*, which functions by aiding in protein folding via chaperoning hydrophobic patches in nascent proteins.
- 5) Observation of tertiary protein structure of PBP2A shows the presence of several beta-pleated sheets and the necessity of chaperone stabilization of nascent PBP2A for proper folding and activity.

The work is elegant in that it displays the relatively "frail" nature of the PBP2A protein in physiological conditions. This group mentions the possibility of targeting the chaperone proteins of PBP2A as an additional "druggable" target against *S. aureus* MRSA. As there are relatively few available antimicrobial drugs at present to combat MRSA (Vancomycin, Linezolid, Ceftaroline), additional targetable systems in MRSA should be exploited.

The rigor of the work appears sound, with adequate replication performed. One question I have for the authors is whether they have considered testing these experiments in other strains of MRSA besides COL? MRSA N315, for example?

Reviewer #3 (Remarks to the Author):

This study uses biochemical and genetic approaches to investigate the post-translocational folding of the resistance determinate PBP2a in MRSA. Specifically, the manuscript demonstrates discrete roles for, and synergy between, the serine protease HtrA1 and the PPIase PrsA in the functional folding of PBP2a and resultant β -lactam resistance.

The study has been very well-performed, the experimental work is extensive with robust controls throughout and supported by comprehensive supplemental material. Likewise, the manuscript is very well-written with excellent clarity and detail. Together, this makes for an impressive study, offering novel data of considerable interest to a wide range of readers.

I have the following specific comments:

1. The authors' provide some structure-function data with regards to HtrA but not PrsA. They do show that PrsA possesses chaperone activity in an unrelated assay but it is not clear if it is this activity and/or PPIase activity which contributes to PBP2a folding and oxacillin resistance. To answer this would provide insightful mechanistic detail to match that presented for HtrA and unless practically unfeasible should be seriously considered.
2. It should be clarified whether or not the PPIase domain of PrsA been shown to confer that particular enzymatic activity?
3. Line 316 correct *Streptococcus pneumoniae*.
4. Figure 2C I suggest in the HtrA schematic that the active site S is replaced by A to make clearer the nature of the mutation rather than simply crossing out the S.
5. Table 1 lower case v in *vraSR/vsaS*

COMMSBIO-19-1015-T Roch M. *et al*
Detailed response to reviewer's comments:

We thank the three anonymous reviewers for their comments and suggestions for improvements to our manuscript. We have responded to all inquiries and the corresponding text changes are highlighted in Word tracking mode in the accompanying uploaded files.

Reviewers' comments:

Reviewer #1 (Remarks to the Author):

In their study Roch et al. use complementary genetic, structural and biochemical approaches to show that PBP2A activity is impacted by a catastrophic folding transition, necessitating active interactions with the extracellular folding factors PrsA and HtrA1 to ensure proper post-translocational PBP2A maturation and sustained MRSA β -lactam resistance. The results presented in this study could inspire novel approaches to counteract β -lactam resistance in MRSA, by targeting the extracellular protein folding machinery.

Overall, the experimental evidence presented in this paper convincingly shows that functional PBP2A maturation is reliant on interactions with folding-factors at physiological conditions following its vectorial transit through the secretion translocation channel. While the authors have previously highlighted the role of PrsA in the extracellular maturation of PBP2a (doi:10.1128/aac.06264-11, doi:10.1128/aac.02333-15), the comprehensive characterization of HtrA1 and PrsA involvement in the extracellular PBP2A maturation presented in this study provides novel insights into this important prerequisite for β -lactam resistance in MRSA. There are, however, some minor points that need to be addressed.

1. The introduction of the extracellular folding machinery (HtrA, PrsA) in the introduction (line 57) as well as in the result section (line 129) is abrupt. An introduction of these factors that is more cohesive with the previous paragraphs would help to more efficiently convey the authors experimental rationale.

We have added additional new text (lines 56-60; 67-68; 137-140) that helps define and clarify the experimental rationale.

2. The authors claim that purified PBP2A was stable at 5mg/ml and 4C for up to five months (lines 97-98). This claim should be supported by the corresponding data or a relevant reference.

We have supported the claim made in the text by supplying additional data in the revised Supplementary materials, Figure 1C. We now show bocillin FL assays performed six months apart (Supplementary Figure 1B, C) with the same aliquot of our purified PBP2A that had never been frozen/thawed, and which had been stored at 4°C. In these storage conditions, the protein did not aggregate, or form visible precipitates. Visible precipitates are clearly evident when performing assays at various temperatures and pH as we show in Figure 1. Thus, long term storage *per se* does not lead to protein inactivation, whereas exposure to elevated temperature or acidic pH conditions as we report clearly does.

3. The results presented in Fig4 (VraTSR regulation of htrA1 and prsA) need more context than that provided at first mention (line 135). The authors provide a comprehensive interpretation of their results later (lines 148-152). As the first mention of VraTSR is solely aimed at establishing its regulatory control over htrA1 and prsA reference of one of the many publications providing evidence for this relationship here might be more appropriate (e.g. Boyle-Vavra Antimicrob Agents Chemother 2013 or their own Antimicrob Agent Chemother publication from 2012).

We have added next text for clarification of this point (lines 154-156) together with new reference citation 29 (Boyle-Vavra 2013) on line 153.

4. References to *VraTSR* should be harmonized throughout the paper. First, the regulatory system is generally referred to as *VraTSR* and not *VraRST* (as in line 135). Second, the authors use the nomenclature *vraTSR* (line 135) and *vraRS* (line 148) interchangeably. For consistency nomenclature should be harmonized.

We have made the change to *VraTSR* for consistency throughout.

5. The authors claim that coordinate expression under cell wall stress conditions is 'strong evidence' for cooperation (lines 136-137). However, a multitude of genes is co-regulated under cell wall stress conditions, not all of which cooperate. Therefore, coordinate expression alone, specifically in response to an external stressor, is not necessarily evident of cooperation and this statement should be rephrased.

We have rephrased the sentence that now occupies lines 153-156.

6. The authors claim that disruption of *VraTSR* 'was noticeably less severe when compared with either single or double *prsA/htrA1* mutation' (lines 148-152). However, the data provided in Table 1 do not convincingly support this claim, but show that at 37°C *COL ΔvraSR* is actually more susceptible to Oxacillin (MIC 64, range 64-128) when compared to *COL ΔhtrA1* (MIC 128, range 128-256) or *COL ΔprsA* (MIC 128). Therefore, the authors claim should be rephrased or removed.

Thank you for this. We have rephrased the sentence to reflect that we were referring to the comparison with the *prsA/htrA1* double mutant (line 171).

7. The authors claim that the dual *htrA1/prsA* operon on plasmid *pCN47* does 'restore resistance to β -lactams' (lines 158-159). While their data support restored resistance against low concentrations of Oxacillin (32mg/ml) the complemented double mutant remains substantially more susceptible to CPT and high levels of OXA (128 mg/ml) when compared to wild type or complemented single mutant strains. While this observation could represent an artefact resulting from the housekeeping promoter used to drive *htrA1/prsA* expression, this fact should be acknowledged in the interpretation of the results.

We have clarified description of our finding in new text lines 178-194.

8. The authors should provide a reference for functional/structural studies that have shown restored growth of hypersensitive *E. coli* strains when complemented with trigger factor (lines 224-225).

We have added an additional point of citation for this reference (now reference 41) that was present in the original manuscript following "25°C." The citation is now on both lines 267 and again on line 270 for clarity.

9. The reference to Fig 4 in line 258 is unnecessary.

We have removed the figure reference as suggested.

10. The statement that '*PrsA* and *HtrA1* together assure the proper maturation of *PBP2A* during its vectorial transit from the secretion translocation channel', should be rephrased as the authors cannot exclude contribution of additional factors during this process.

We have revised this portion of the text in the discussion- lines 319-324.

11. The authors claim that 'Colony viability was remarkably reduced to a level that would predictably compromise even the survival of rare spontaneous mutants conferring altered resistance to ceftaroline'. Moreover, they state that modulating *PBP2A* post-transcriptional folding could 'possibly contribute to enhanced therapeutic efficacy by compromising the emergence of resistance'. For both of statements the authors fail to provide any data that would support their claims. Until now there has not been an evolution-proof antimicrobial therapy, but every drug/drug

combination has eventually been met with the rise of resistant clones. Therefore, these statements should be rephrased or removed.

We have removed the statement in the original text and modified the remaining text- new lines 352-353.

12. The legend of Supplementary Figure 2 does not indicate which part of the legend corresponds to subpanel A or B.

Thank you for pointing this out. We have modified the legend of supplementary figure 2 as requested.

13. Supplementary Figure 3B should include the MIC data of the wild type strain for context.

We have included the MIC data for the wild type strain as requested.

Language suggestions:

Line 127 – Remove ‘therefore’.

Line 160 – Remove the double spacing at beginning of sentence.

Line 317 – ‘Thus, the dependence of the critical folding of important ... for β -lactam resistance upon extracellular...’ instead of ‘Thus, the critical folding ... for β -lactam resistance dependent upon extracellular ...’.

We have incorporated all suggestions: now line 137, 195, 363-365.

Reviewer #2 (Remarks to the Author):

*Roch and colleagues from the Kelley Group further investigated the possibilities of studying chaperone proteins in *S. aureus* MRSA for biochemical frailties in the PBP2A protein.*

Their work in this manuscript described several experiments:

- 1) Perturbations of PBP2A 3D structure using physiologically significant modulations of heat and pH.*
- 2) Effects on beta-lactam resistance via perturbations of PrsA and HtrA1 proteins and their roles as chaperones in proper folding and anti-aggregation in MRSA COL. Complementation with plasmids for these proteins validated their roles in beta-lactam resistance. The use of Ceftriaxone (latest-gen cephalosporin with anti-PBP2A activity) and Oxacillin (earlier non-anti-PBP2A penicillin) were appropriate test agents for beta-lactam drugs in MRSA.*
- 3) Use of Western blot analysis to confirm aggregation of misfolded PBP2A without functional HtrA1.*
- 4) PrsA protein appears to have homologous function in *S. aureus* to Trigger factor in *E. coli*, which functions by aiding in protein folding via chaperoning hydrophobic patches in nascent proteins.*
- 5) Observation of tertiary protein structure of PBP2A shows the presence of several beta-pleated sheets and the necessity of chaperone stabilization of nascent PBP2A for proper folding and activity.*

*The work is elegant in that it displays the relatively "frail" nature of the PBP2A protein in physiological conditions. This group mentions the possibility of targeting the chaperone proteins of PBP2A as an additional "druggable" target against *S. aureus* MRSA. As there are relatively few available antimicrobial drugs at present to combat MRSA (Vancomycin, Linezolid, Ceftriaxone), additional targetable systems in MRSA should be exploited.*

The rigor of the work appears sound, with adequate replication performed. One question I have for the authors is whether they have considered testing these experiments in other strains of MRSA besides COL? MRSA N315, for example?

The single mutations in *prsA* have been made in MRSA strains Mu50, Mu3, and MW2 as published in previous work by Jousselin et al 2012 and 2015 in our laboratory (references herein 21, 22) and they indeed detectably affect oxacillin resistance. Data for this was shown in these manuscripts. We have made *htrA1* mutation in Mu3. We have considerable difficulty making double *prsA/htrA1* mutation in alternative strains to COL, since during the course of bacteriophage transduction, strains consistently deleted the *mec* element. Even the COL double mutations were difficult to obtain and as noted in our methods, we always checked by PCR assay for *mecA* retention. These findings suggest to us that the loss of *prsA/htrA1* in these other MRSA backgrounds imposes a considerable fitness cost. There are follow-up studies planned with these strains to understand this, but we first must solve the technical challenge of genetic construction. Since COL has been the gold standard for decades of MRSA research, we elected to proceed with this strain where we were able to obtain all mutation combinations.

Reviewer #3 (Remarks to the Author):

This study uses biochemical and genetic approaches to investigate the post-translocational folding of the resistance determinate PBP2a in MRSA. Specifically, the manuscript demonstrates discrete roles for, and synergy between, the serine protease HtrA1 and the PPIase PrsA in the functional folding of PBP2a and resultant β -lactam resistance.

The study has been very well-performed, the experimental work is extensive with robust controls throughout and supported by comprehensive supplemental material. Likewise, the manuscript is very well-written with excellent clarity and detail. Together, this makes for an impressive study, offering novel data of considerable interest to a wide range of readers.

I have the following specific comments:

1. The authors' provide some structure-function data with regards to HtrA but not PrsA. They do show that PrsA possesses chaperone activity in an unrelated assay but it is not clear if it is this activity and/or PPIase activity which contributes to PBP2a folding and oxacillin resistance. To answer this would provide insightful mechanistic detail to match that presented for HtrA and unless practically unfeasible should be seriously considered.

We performed a structure/function study with PrsA in our previously published work Jousselin et al 2015 (reference 22 herein). In this published study, we found that we could complement a *prsA* disruption for oxacillin resistance (up to 500 μ g/ml in colony forming agar plate assay) with either the full length *prsA*, or *prsA* lacking the PPIase parvulin domain (a construct called NC in that work indicating N- and C-terminal domains). Thus, in this context (as cited) and with our reported current results with PrsA functionally substituting for *E. coli* trigger factor, with or without its parvulin domain, we feel that the chaperone activity that PrsA provides is most likely the anti-aggregation activity (contributed by the N- and C- domains). We cannot exclude that the PrsA PPIase contributes in some way, but it would appear to be a minor contribution, if at all.

2. It should be clarified whether or not the PPIase domain of PrsA been shown to confer that particular enzymatic activity?

***S. aureus* PrsA parvulin domain is indeed a PPIase by biochemical assay, and this was reported in a structural study Heikkinen O et al (2009) *BMC Structural Biology*. Reference to this fact and this particular reference cited was made in our previously published study, Jousselin et al (2015) that we cite herein.**

3. Line 316 correct Streptococcus pneumoniae.

We have made the correction- line 363.

4. Figure 2C I suggest in the HtrA schematic that the active site S is replaced by A to make clearer the nature of the mutation rather than simply crossing out the S.

We have modified Figure 2C. Although we have retained the X to draw attention to the site of mutation, we have added to the right of the schematic in Fig2C explicit description of each construct: *htrA1+*, *htrA1 S255A*, and *htrA1 ΔPDZ*.

5. Table 1 lower case v in vraSR/vsaS

We have made the proper nomenclature changes in Table 1, and Supplementary Figure 4 also.